# Detection of Secondary Side Position for Segmented Dynamic Wireless Charging Systems Based on Primary Phase Angle Sensing

**Wei Xiong** [1,2]**, Jiangtao Liu** [1,2]**, Jing Chen** [3,]*** and Dewang Hu** [3]

1   School of Physics and Mechanical and Electrical Engineering, Hubei University of Education,
    Wuhan 430205, China
2   Expert Workstation for Terahertz Technology and New Energy Materials and Devices, Wuhan 430205, China
3   Wuhan Second Ship Design and Research Institute, Wuhan 430225, China
*   Correspondence: drchen@whu.edu.cn

**Abstract:** In dynamic wireless charging systems, the detection of secondary side positions has been attracting much attention in academic research. Due to the strong electromagnetic interference and the presence of foreign objects in the charging area, the use of conventional detection methods such as wireless communication and infrared techniques may be problematic; therefore, as an alternative to solve the above problem, a new detection method based on phase angle sensing is proposed in this paper. Through phase analysis of the primary input impedance and by reference to the relationship between the input port phase angle and the secondary side position, the proposed method is able to sense the secondary side position in real time. In addition, an analysis of the sensitivity of the proposed method to parameter variations is also carried out. In order to verify the effectiveness of the proposed position detection method, a dynamic wireless charging system with four segments is built for experimental verification. The experimental results show that when the phase angle threshold is set at $300°$, the secondary side position can be accurately identified, and the proposed method is quite robust within a parameter deviation of up to 4%.

**Keywords:** electric vehicle; dynamic wireless charging; secondary side position detection; phase angle sensing; sensitivity analysis

## 1. Introduction

Wireless Power Transfer (WPT) has the advantages of safety, convenience, waterproofing, dustproofing, etc. It has been widely used in the field of electric vehicles (EV), consumer electronics, etc. [1–6].

To ensure a certain endurance mileage, electric vehicles are usually equipped with high-capacity batteries. However, the heavy and high-capacity battery pack not only increases the weight of the electric vehicle but also increases the energy consumption of the electric vehicle during driving. Dynamic WPT [7,8] allows electric vehicles to obtain electrical energy wirelessly while driving, thus reducing the dependence of electric vehicles on batteries [9].

A segmented dynamic WPT system refers to a dynamic WPT system containing several charging areas. In the segmented dynamic WPT system, when the electric vehicle enters the corresponding wireless charging area, the power supply of the charging coil in the area shall be turned on to supplement electrical energy for the electric vehicle in wireless mode. In general, in the dynamic charging area, sensors can be used to detect the corresponding position of the electric vehicle [10–12]. On one hand, the sensor setting increases the manufacturing cost of the system; on the other hand, it also has the problem of foreign-matter blocking, which reduces the reliability of the system. At the same time, in highway applications, the traffic flow is large and the sensor is difficult to install and

detect. Reference [13] introduced a three-coil position detection system, which designed two detection coils on the primary side (charging side) of the wireless charging system and one detection coil on the secondary side (EV side). The three-coil detection system can normally identify when a single electric vehicle gradually enters the two detection coils; however, the system has difficulty in accurately identifying if multiple electric vehicles enter the two detection coils at the same time. In other words, when electric vehicles come from other lanes rather than the same lane, the system cannot accurately detect whether electric vehicles are approaching this lane. Reference [14] proposed a control algorithm to adjust the wireless charging power of the dynamic wireless charging system without using other position-sensing devices. However, the transmitter side coil of the system is always working, and the literature does not consider the system standby loss under this condition.

This paper proposes a position detection scheme for electric vehicle dynamic WPT systems in sections based on real-time phase angle perception, which eliminates the need for complex schemes such as communication and external infrared sensors. By solely detecting the internal circuit parameters, the position of secondary side vehicles can be identified in real time. The overall structure of the system is shown in Figure 1. The system includes $n$ dynamic charging areas. Each charging area contains a wireless charging transmitting coil with a self-inductance $L_i$. Each transmitting coil is connected to a high-power AC power supply $P_i$ and a low-power AC power supply $S_i$. Both AC sources have the function of sensing and measuring the impedance phase angle ($\varphi_{PAVR}$) of the resonator port, which can be used to detect whether the electric vehicle enters or leaves the charging area. The electric vehicle is equipped with a receiving coil with a self-inductance, $L_s$. When there is no electric vehicle in the charging area, the transmitting coil is only connected to the low-power AC source, and the high-power AC source is turned off to save electrical energy; when a vehicle entering the charging area is detected, the low-power AC source is turned off and the high-power AC source is turned on to charge the electric vehicle dynamically and wirelessly.

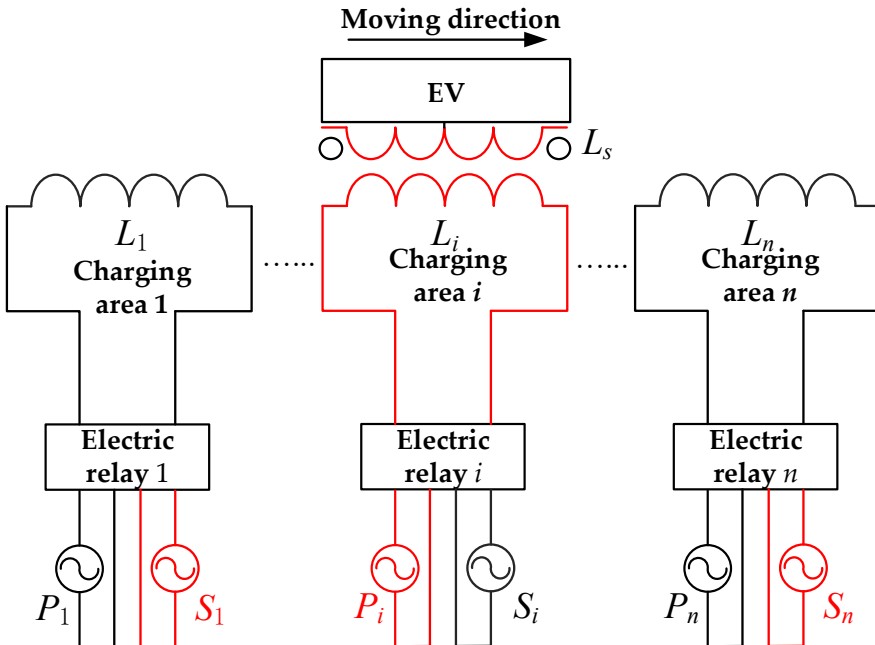

**Figure 1.** Schematic diagram of the system structure.

## 2. Analysis of the Impedance Phase Angle Model of the Resonant Port

The equivalent circuit of the electric vehicle dynamic wireless charging model can be simplified as shown in Figure 2.

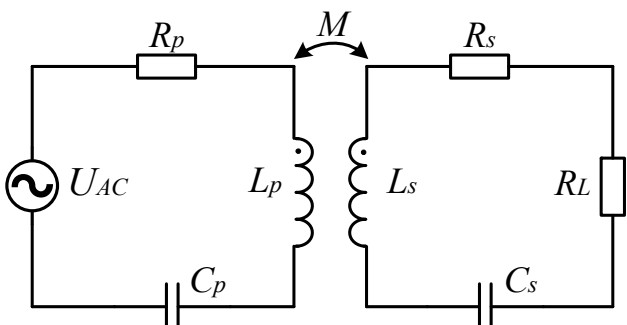

**Figure 2.** Equivalence circuit diagram.

As can be seen from Figure 2, $U_{AC}$ is the AC source, $R_P$, $L_P$, and $C_P$ are the inner resistance, self-inductance of the transmitting coil, and compensation capacitance, respectively, $R_S$, $L_S$, and $C_S$ are the inner resistance, self-inductance of the receiving coil and compensation capacitance, respectively, $R_L$ is the load, and $M$ is the mutual inductance between the transmitting and receiving coils. Then, the frequency domain equation of the system can be established according to the classical circuit theoretical model. In the primary circuit and secondary circuit there are:

$$Z_P = R_P + j\omega L_P + \frac{1}{j\omega C_P} \tag{1}$$

$$Z_S = R_S + R_L + j\omega L_S + \frac{1}{j\omega C_S} \tag{2}$$

where $Z_{PS}$ is the reflected impedance from the secondary circuit to the primary circuit, which can be calculated as [15]:

$$Z_{PS} = \frac{(\omega M)^2}{Z_S} \tag{3}$$

$$Z_P = R_P + j\omega L_P + \frac{1}{j\omega C_P} + Z_{PS} \tag{4}$$

The resonant angle frequency of the transmitting end in all power supply areas is $\omega_{P0}$, and the resonant angular frequency of the receiving end is $\omega_{S0}$, so that the quality factors $Q_P$ and $Q_S$ of the transmitting end and the receiving end can be expressed as:

$$Q_P = \frac{\omega_{P0} L_P}{R_P} = \frac{1}{\omega_{P0} C_P R_P} \tag{5}$$

$$Q_S = \frac{\omega_{S0} L_S}{R_S} = \frac{1}{\omega_{S0} C_S R_S} \tag{6}$$

The deviation between the operating frequency of the primary circuit and the secondary circuit is set and the resonant frequency is $F_P$ and $F_S$. Then, there are:

$$F_P = \frac{\omega}{\omega_{P0}} - \frac{\omega_{P0}}{\omega} \tag{7}$$

$$F_S = \frac{\omega}{\omega_{S0}} - \frac{\omega_{S0}}{\omega} \tag{8}$$

where $\omega$ is the angular frequency. Substitute Formulas (2), (3), (5) and (6) into Formula (4) to obtain:

$$Z_P = R_P + \frac{(\omega M)^2}{(R_S + R_L)(1 + (F_S Q_S)^2)} + j(R_P F_P Q_P - \frac{F_S Q_S(\omega M)^2}{(R_S + R_L)(1 + (F_S Q_S)^2)}) \quad (9)$$

Then, the impedance phase angle $\varphi_{PAVR}$ of the resonator port of the primary circuit can be calculated using (9):

$$\tan(\varphi_{PAVR}) = \frac{\text{Imag}(Z_P)}{\text{Real}(Z_P)} = \frac{F_P Q_P(1 + (F_S Q_S)^2) - \frac{Q_P F_S Q_S^2 k^2 \omega^2}{\omega_{P0}\omega_{S0}}}{1 + (F_S Q_S)^2 + \frac{Q_P Q_S k^2 \omega^2}{\omega_{P0}\omega_{S0}}} \quad (10)$$

According to Formula (10), for a given system, $\varphi_{PAVR}$ varies with the change in coupling coefficient $k$.

## 3. Analysis of the Model between Coupling Coefficient and Vehicle Position

When the electric vehicle is running in the power supply area, because the transmitting coil is far larger than the receiving coil, the transmitting coil can be approximately regarded as a rectangle in a short distance. The relationship between the receiving coil and the transmitting coil positions is shown in Figure 3.

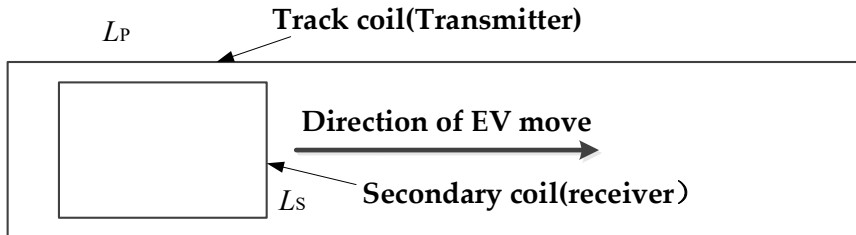

**Figure 3.** Swatch diagram for the electric vehicle receiving coil and the transmitting coil relative positions.

When the electric vehicle runs to the power supply area of a transmitting coil, the transmitting coil and the receiving coil form a coaxial two-coil model along the driving direction of the electric vehicle. The model diagram is shown in Figure 4.

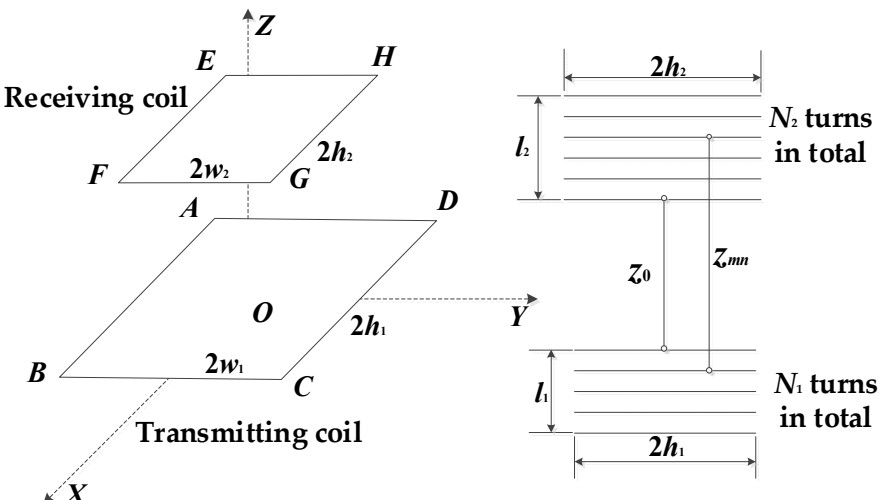

**Figure 4.** Double coil model schematic diagram.

As can be seen from Figure 4, the side lengths of the transmitting coil are $2w_1$ and $2h_1$, respectively, with $N_1$ turns in total, and the side lengths of the receiving coil are $2w_2$ and $2h_2$, respectively, with $N_2$ turns in total. Then, the mutual inductance of the transmitting coil and receiving coil can be expressed as [16,17]:

$$M = \sum_{m=1}^{N_1} \sum_{n=1}^{N_2} \left( \Phi_{(AB-Z)}(i,j) + \Phi_{(BC-Z)}(i,j) + \Phi_{(CD-Z)}(i,j) + \Phi_{(DA-Z)}(i,j) \right) \tag{11}$$

where $\Phi_{(AB-Z)}(i,j)$ represents the magnetic flux generated on the $j$th turn of the receiving coil along the Z-axis direction when a current of 1A effective value flows on the AB side of the $i$th turn of the transmitting coil. $\Phi_{(BC-Z)}(i,j)$, $\Phi_{(CD-Z)}(i,j)$, and $\Phi_{(DA-Z)}(i,j)$ are the same.

A mathematical model is established and a group of parameters are set for simulation calculation to simulate the actual situation of electric vehicles. In the simulation model, the side length of the transmitting coil is $2w_1 = 650$ mm, $2h_1 = 76$ mm, $l_1 = 16$ mm, and turns $N_1 = 10$; the side length of the receiving coil is $2w_2 = 152$ mm, $2h_2 = 46$ mm, $l_2 = 16$ mm, and the number of turns is $N_2 = 20$. The distance between the center points of the two coils $Z_0 = 15$ mm, and the relative displacement of the receiving and transmitting coils when the electric vehicle is actually running is simulated. The relative displacement of the two coils is set as –600 mm to 600 mm. The mutual inductance of the two coils can be calculated according to Formula (11).

When the width, height, length, and turns of the rectangular coil, i.e., $w$, $h$, $l$, and $N$ are determined, the inductance can be calculated using the Niwa formula given by Grover [18], as follows:

$$
\begin{aligned}
L = {} & 0.008 N_t^2 \frac{\mu_r wh}{l} \Big[ \Big( \frac{\pi}{2} - \tan^{-1} \frac{wh}{l^2 \sqrt{1 + g^2/l^2}} \Big) + \frac{1}{2} \Big( \frac{l}{h} \sinh^{-1} \frac{w}{h} \Big) + \\
& \frac{1}{2} \Big( \frac{l}{w} \sinh^{-1} \frac{h}{l} \Big) - \frac{1}{2} \Big( \Big( 1 - \frac{h^2}{l^2} \Big) \frac{l}{h} \sinh^{-1} \frac{w}{l\sqrt{1 + h^2/l^2}} \Big) \\
& - \frac{1}{2} \Big( \Big( 1 - \frac{w^2}{l^2} \Big) \frac{l}{w} \sinh^{-1} \frac{h}{l\sqrt{1 + w^2/l^2}} \Big) - \frac{1}{2} \Big( \frac{h}{l} \sinh^{-1} \frac{w}{h} \Big) \\
& - \frac{1}{2} \Big( \frac{w}{l} \sinh^{-1} \frac{h}{w} \Big) + \frac{1}{3} \Big( \frac{l^2}{wh} \Big( 1 - \frac{g^2}{2l^2} \Big) \sqrt{1 + \frac{g^2}{l^2}} \Big) - \frac{1}{3} \Big( \frac{l^2}{wh} \Big( 1 - \frac{w^2}{2l^2} \Big) \sqrt{1 + \frac{w^2}{l^2}} \Big) \\
& - \frac{1}{3} \Big( \frac{l^2}{wh} \Big( 1 - \frac{h^2}{2l^2} \Big) \sqrt{1 + \frac{h^2}{l^2}} \Big) + \frac{1}{6} \Big( \frac{g^3 - w^3 - h^3}{whl} \Big) + \frac{l^2}{3wh} \Big]
\end{aligned}
\tag{12}
$$

Therefore, the inductance of the transmitting and receiving coils can be calculated using Formula (12), where $L_P$ is 191.82 μH and $L_S$ is 64.21 μH. At the same time, a coupling coefficient calculation formula is given, as follows:

$$k = \frac{M}{\sqrt{L_S L_P}} \tag{13}$$

When there is a relative displacement between the receiving and transmitting coils, the change in the coupling coefficient $k$ can be calculated, as shown in Figure 5. It can be seen from Figure 5 that when the receiving and transmitting coils have relative displacement, the coupling coefficient quickly reaches the peak value when the axis of the receiving coil just enters the edge of the transmitting coil, and then with the movement of the receiving coil, the coupling coefficient decreases slightly and remains at a high level until it reaches the edge of the other side. Therefore, in this system, when the electric vehicle operates normally in the power supply area, the coupling coefficient has a large range of changes, which will be reflected in the impedance phase angle of the primary resonant cavity port. Therefore, the secondary coil can be positioned through the real-time sensing technology of the phase angle.

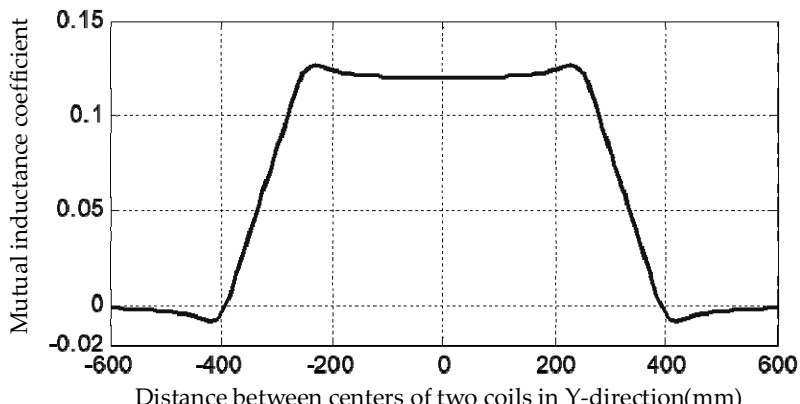

**Figure 5.** Receive the interdependence coefficient of the transmitting coil.

## 4. Analysis of Impedance Phase Angle Sensing Scheme of Parameter Sensitivity and Robustness

It can be seen from Formula (10) that when the system operates at the resonant frequency, i.e., $F_P = F_S = 0$, $\varphi_{PAVR}$ is 0. Obviously, $\varphi_{PAVR}$ cannot be used to detect the position of the electric vehicle on the track. When the operating frequency of the system is not equal to the resonant frequency, $\varphi_{PAVR}$ will change with the change in the coupling coefficient $k$. When the variation range of the phase angle $\varphi_{PAVR}$ is large enough, the system can obtain a better judgment condition. Therefore, this section will focus on how to make the variation range of $\varphi_{PAVR}$ larger.

### 4.1. Simulation of $\varphi_{PAVR}$ Varying with System Frequency and Coupling Coefficient k

The following parameters in Table 1 are used for Matlab simulation to analyze the influence of operating frequency and electric vehicle position on $\varphi_{PAVR}$, as shown in Figure 6. In the figure, when the operating frequency varies from 1 to 1.08 times the resonant frequency, the $\varphi_{PAVR}$ increases first and then decreases. Moreover, when the electric vehicle enters the edge of the track coil, $\varphi_{PAVR}$ will decrease sharply, and when the electric vehicle leaves, $\varphi_{PAVR}$ will increase rapidly.

**Table 1.** Matlab simulation parameters.

| Parameter | Attribute | Value |
| --- | --- | --- |
| $f_{res}$ | resonant frequency | 85 kHz |
| $R_P$ | resistance of the transmitter coil | 0.5 Ω |
| $L_P$ | coil inductance at the transmitter | 191.82 μH |
| $C_P$ | compensation capacitance at the transmitter | 18.28 nF |
| $R_S$ | resistance of receiver coil | 0.5 Ω |
| $L_S$ | coil inductance at the receiver | 64.21 μH |
| $C_S$ | compensation capacitance at the receiver | 54.61 nF |
| $R_L$ | load | 10 Ω |

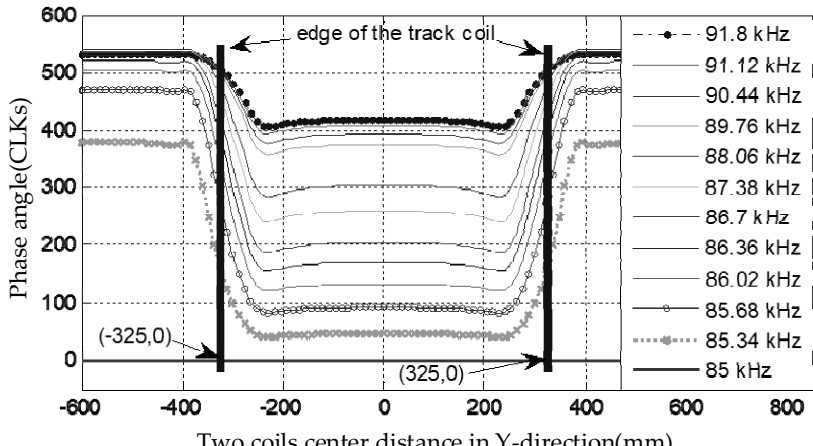

**Figure 6.** Frequent angle changes in the location and working frequency of the vehicle.

In order to accurately analyze the performance of the system when detecting the position of the electric vehicle, the variation range of the phase angle, $\varphi_{PAVR}$, is taken as a reference. When the two coils are coaxial and the coupling coefficient is $k$, Formula (10) can be expressed as:

$$\varphi_{PAVR} = \arctan(F_P Q_P) - \arctan\left(\frac{F_P Q_P (1 + (F_S Q_S)^2 - \frac{Q_P F_S Q_S^2 k^2 \omega}{\omega_{P0} \omega_{S0}})}{(1 + (F_S Q_S)^2 + \frac{Q_P Q_S k^2 \omega^2}{\omega_{P0} \omega_{S0}})}\right) \tag{14}$$

Figure 7 shows the curve of $\varphi_{PAVR}$ changing with the ratio of working frequency to resonant frequency when the parameter setting is consistent with Figure 6. It can be seen from the figure that when the ratio of operating frequency to resonant frequency exceeds 1.04, $\varphi_{PAVR}$ is lower than 200. Considering the error of components in the hardware circuit, and in order to obtain better zero voltage switching conditions, the ratio of operating frequency to resonant frequency is usually higher than 1.05. In Figure 7, when the ratio of operating frequency to resonant frequency is high, $\varphi_{PAVR}$ is small, which is not conducive to the measurement of $\varphi_{PAVR}$. Therefore, corresponding measures should be taken to improve $\varphi_{PAVR}$ to improve the accuracy of measurement.

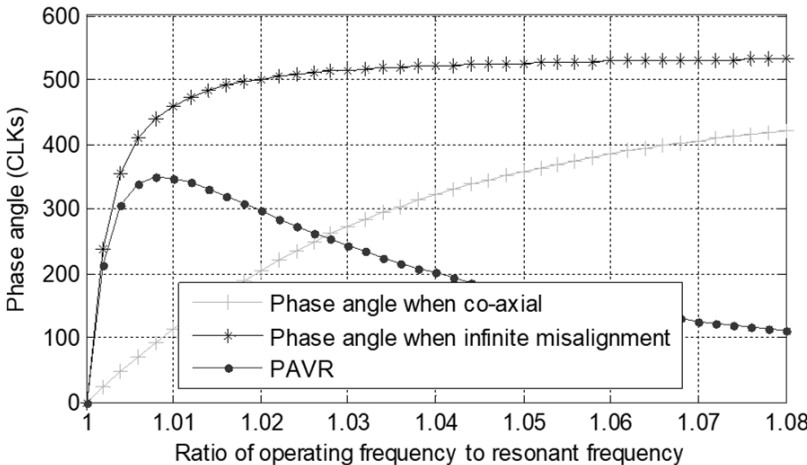

**Figure 7.** Changes in $\varphi_{PAVR}$ with changes in the ratio of work and resonance frequency.

It can be seen from Formula (14) that $\varphi_{PAVR}$ is mainly affected by $Q_P$, $Q_S$, $k$, and operating frequency f. However, $Q_P$, $Q_S$, and $k$ are independent of each other, so $\varphi_{PAVR}$ can be improved by optimizing these parameters. When $k$ is 1.5 times the parameters

in Figure 6, $Q_S$ is 1.5 times, and $Q_P$ is 0.67, $\varphi_{PAVR}$, in Figure 8, is significantly improved compared to $\varphi_{PAVR}$ in Figure 7.

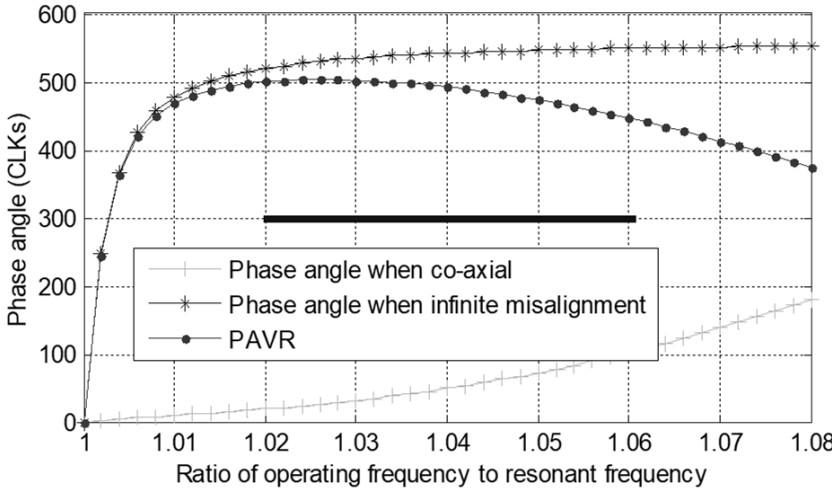

**Figure 8.** The simulation diagram for $\varphi_{PAVR}$ with the ratio of work to the resonance frequency.

### 4.2. System Robustness

Due to the tolerance and parameter drift of the components, the resonant frequencies of the transmitter and receiver may shift to some extent. Therefore, we change the operating frequency to discuss the robustness of the system. The robustness of the system is also affected by the ratio of the selected operating frequency to the resonant frequency. In fact, the appropriate ratio for the selected operating frequency to the resonant frequency should be a compromise between robustness and $\varphi_{PAVR}$. That is, when $\varphi_{PAVR}$ is large enough, the operating frequency has a large bandwidth. In Figure 8, although the maximum value of $\varphi_{PAVR}$ appears when the ratio of the operating frequency to the resonant frequency is 1.02, the operating frequency is too close to the resonant frequency. When the actual resonant frequency is slightly larger due to the parameter drift of the components, $\varphi_{PAVR}$ will decline significantly. If the ratio of the selected operating frequency to the resonant frequency is large, $\varphi_{PAVR}$ will be smaller, which cannot meet the requirements of the coil switching logic. When considering that the operating frequency is far away from the resonant frequency and $\varphi_{PAVR}$ needs to have a medium value, a ratio of 1.04 between the operating frequency and the resonant frequency is a good choice. In addition, when the ratio of the operating frequency to the resonant frequency changes from 1.02 to 1.06, the phase angle difference of the coaxial state and infinite distance can exceed 300°, meeting the requirements of the coil switching logic. Therefore, when the ratio of the operating frequency and the resonant frequency of the system can be selected as 1.04, a large operating frequency bandwidth can be guaranteed while meeting the $\varphi_{PAVR}$ demand, that is, better robustness can be obtained.

## 5. Experimental Verification

### 5.1. Construction of Test Bench

According to the SAE international standard for electric vehicle wireless charging, the resonant frequency of the coil at the transmitting and receiving side of the system design is 85 kHz. According to the above simulation parameters and considering that the electric vehicle track is circular, the electric vehicle track is made as shown in Figure 9, where the coil spacing S is 800 mm, while the length and width of the coil are 630 mm and 100 mm, respectively. The ratio of the operating frequency to the resonant frequency of the system is selected as 1.04, that is, the operating frequency is 88.4 kHz. The main parameters for an electric vehicle rail wireless power supply design and manufacture are shown in Table 2, and 300° is selected as the switching threshold of the system power supply area

during operation. The relevant parameters are measured using an E4980ALCR analyzer manufactured in Germany.

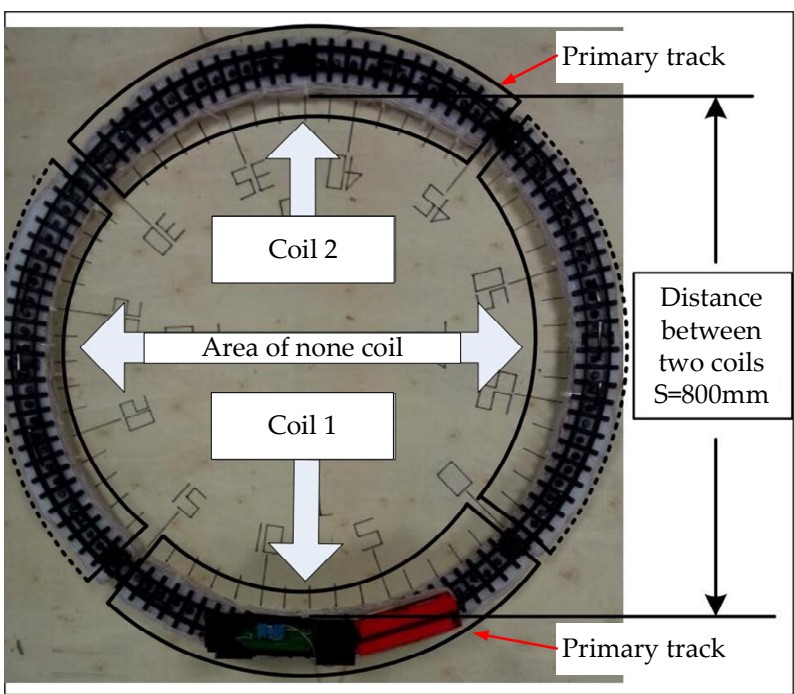

**Figure 9.** Electric vehicle track model.

**Table 2.** System main parameters.

| Parameter | Attribute | Value |
|---|---|---|
| $L_1$ | coil inductance value of NO.1 transmitter | 186.7 μH |
| $C_1$ | compensation capacitance value of NO.1 transmitter | 18.76 nF |
| $R_1$ | resistance of NO.1 transmitter coil | 0.642 Ω |
| $f_1$ | resonant frequency value of NO.1 transmitter | 85.0 kHz |
| $L_2$ | coil inductance value of NO.2 transmitter | 194.73 μH |
| $C_2$ | compensation capacitance value of NO.2 transmitter | 19.76 nF |
| $R_2$ | resistance of NO.2 transmitter coil | 0.612 Ω |
| $f_2$ | resonant frequency value of NO.2 transmitter | 85.0 kHz |
| $L_S$ | inductance value of receiving coil | 81.41 μH |
| $C_S$ | compensation capacitance value of receiving coil | 43.99 nF |
| $R_S$ | load resistance value of receiving coil | 10.27 Ω |
| $R_{ESR}$ | resistance of receiving coil | 0.51 Ω |
| $f_S$ | resonant frequency value of receiving coil | 85.0 kHz |
| $N_P$ | number of turns—transmitter | 15 |
| $N_S$ | number of turns—receiver | 18 |
| $f$ | system working frequency | 88.4 kHz |
| $M_1$ | mutual inductance between NO.1 transmitting and receiving when coaxial | 20.75 μH |
| $M_2$ | mutual inductance between NO.2 transmitting and receiving when coaxial | 20.80 μH |
| $C$ | supercapacitance value of energy storage module | 2.5 F |

As can be seen from Figure 9, the whole track is divided into 60 equal parts and marked with corresponding figures to measure the phase angle when the electric vehicle is running, wherein, the transmitting coils are uniformly distributed from 0 to 15 and from

30 to 45, and no coils are distributed in other areas. When the electric vehicle runs to the area without coils, it is powered by supercapacitor *C*, whose value is shown in Table 2. When the vehicle moves on the transmitter coil 1 or 2, the supercapacitor *C* is charged. The distance between the two coils is about S = 800 mm, and the center distance between the two coils is about S = 876 mm.

*5.2. Experimental Verification*

At the beginning of the experiment, the electric vehicle was pushed into the power supply area. When the system detects that the electric vehicle has been driven into the edge of the coil, it turns off the inverter of the small power signal circuit and turns on the inverter of the large power supply. At the same time, the relay acts to disconnect the coil from the small power signal circuit and turn on the large power circuit. The coil on the transmitting side starts to provide wireless power to the electric vehicle. The receiving terminal circuit carried by the electric vehicle first charges the capacitor module. When the voltage of the capacitor module is sufficient to drive the electric vehicle, the electric vehicle gradually starts to run on the track. When the electric vehicle runs to the non-power supply area, the capacitor module will continue to provide electrical energy to drive the electric vehicle. When the electric vehicle runs to the power supply area again, the transmitting side coil will provide wireless power for the electric vehicle to drive the electric vehicle and charge the capacitor module. So far, the electric vehicle will circulate in each power supply area on the track to realize a dynamic wireless power supply.

When the electric vehicle is running along the track with the No. 1 transmitting coil, the measured $\varphi_{PAVR}$ is shown in Figure 10. In the figure, the actual measured $\varphi_{PAVR}$ is compared with the simulation results. Although there is a certain deviation between the actual value and the simulation value, the $\varphi_{PAVR}$ change trend is generally consistent, especially when the coil is switched, when the curve of the actual value is generally consistent with the simulation value. At the same time, it can be seen from the figure that when the electric vehicle is running between coordinates 1 and 14, coil 1 is connected to a high-power power supply, and coil 1, at other positions, is connected to a low-power signal circuit, that is, the switch from a low-power signal circuit to a high-power circuit occurs between 0 and 1; on the contrary, the switch from a high-power circuit to a low-power signal circuit occurs between 14 and 15. Considering that 0 and 15 are the edge of the coil, this experiment proves that the method for measuring $\varphi_{PAVR}$ can accurately detect whether the electric vehicle enters (i.e., between 0 and 1) or leaves (i.e., between 14 and 15) the charging area.

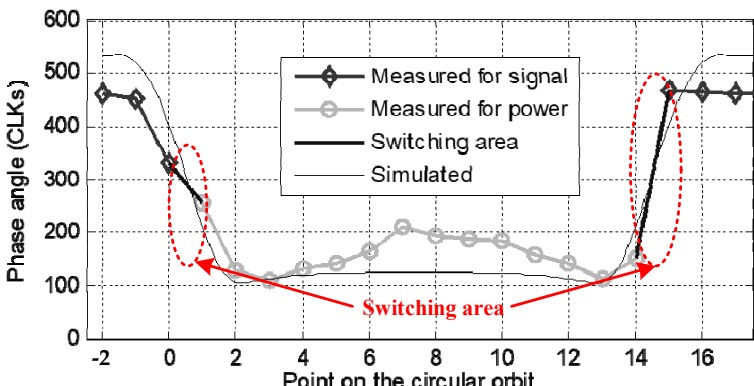

**Figure 10.** The measured $\varphi_{PAVR}$ when the EV moves along the track with the No. 1 transmitting coil.

Figure 11 shows the current of the primary segment 1 while the vehicle moves along the track. The green line shows the turn-on time of the segment, and the position is between the circular orbit 0–1. The red line shows the turn-off time of the segment, and the position is between the circular orbits 14–15.

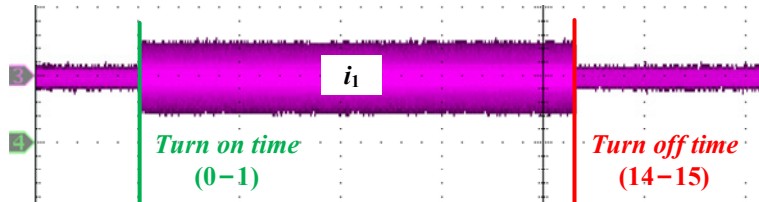

**Figure 11.** The current of primary segment 1 during the vehicle moves along the track.

To verify the robustness of the system when parameter drift occurs, the resonant frequency of the transmitter coil is adjusted to 0.98 times and 1.02 times the original resonant frequency of the experiment, that is, the resonant frequency of the transmitter is now 85 kHz × 98% = 83.2 kHz and 85 kHz × 102% = 86.7 kHz. The resonant frequency is equivalent to a maximum 4% parameter deviation in capacitance or inductance. When other parameters remain unchanged, the relative capacitance and inductance parameters are shown in Table 3.

**Table 3.** The capacitance induction parameters after resonant frequency adjustment.

| Parameter | Attribute | Value |
|---|---|---|
| $L_1$ | coil inductance value of NO.1 transmitter | 184.18 µH<br>185.80 µH |
| $C_1$ | compensation capacitance value of NO.1 transmitter | 19.88 nF<br>18.14 nF |
| $f_1$ | resonant frequency value of NO.1 transmitter | 83.2 kHz<br>86.7 kHz |
| $L_2$ | coil inductance value of NO.2 transmitter | 194.56 µH<br>195.49 µH |
| $C_2$ | compensation capacitance value of NO.2 transmitter | 18.76 nF<br>17.24 nF |
| $f_2$ | resonant frequency value of NO.2 transmitter | 83.4 kHz<br>86.7 kHz |

Figure 12 shows the variation in $\varphi_{PAVR}$ when the electric vehicle is running along the track with the No. 1 transmitting coil under two different resonant frequencies. As can be seen from Figure 12, when the resonant frequency is 83.2 kHz, the switching coordinates are between 0 and 1 and between 13 and 14; when the resonant frequency is 86.7 kHz, the switching coordinates of the coil are between –1 and 0 and between 15 and 16. The experiment shows that the method can still detect the position of an electric vehicle when the parameters drift.

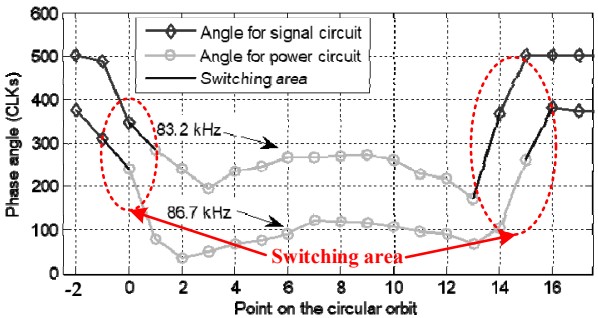

**Figure 12.** The measured $\varphi_{PAVR}$ when the EV moves along the track with the No. 1 transmitting coil under two different frequencies.

## 6. Conclusions

This paper introduces a method for detecting the dynamic wireless charging position of electric vehicles in sections based on the real-time perception of impedance phase angle. By detecting the current lag behind the voltage phase angle in the transmitting coil, this method can detect whether the dynamic wireless charging electric vehicle enters or leaves the charging area, so as to control whether the charging coil in the power supply area is turned on or off according to the electric vehicle position. The experiment shows that the position detection system can accurately identify the secondary side position, and turn on the primary side coil under the premise that the phase angle threshold is $300°$. In addition, the proposed method still has good robustness with a parameter deviation of up to 4%.

**Author Contributions:** W.X. and J.L. conceived and designed the experiments; W.X. and J.C. performed the experiments; W.X. and J.C. wrote the paper; J.L. and D.H. analyzed the experimental results. All authors have read and agreed to the published version of the manuscript.

**Funding:** This work was supported in part by the National Natural Science Foundation of China under Grant 51907054.

**Data Availability Statement:** The data is unavailable due to privacy restriction.

**Conflicts of Interest:** The authors declare no conflict of interest.

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
