# Peer review of "Detection of Secondary Side Position for Segmented Dynamic Wireless Charging Systems Based on Primary Phase Angle Sensing"

_electronics, doi:10.3390/electronics12092148_

Round 1

Reviewer 1 Report

The authors must attend to the following observations

Expand the experimental tests that allow validation of the proposal made for the detection of the secondary side of the wireless charger of an electric vehicle

The conclusions must be validated with more experimental results.

Author Response

Dear Reviewer,

        The authors would like to thank the encouraging and insightful comments. The manuscript has been revised, considering all the comments. The salient additions to the paper were highlighted in red.

  • Reviewer: #1

(1) The authors must attend to the following observations. Expand the experimental tests that allow validation of the proposal made for the detection of the secondary side of the wireless charger of an electric vehicle. The conclusions must be validated with more experimental results.

Response: Thank you for your comment. Now in the revised paper, we have added more experimental results.

Figure 11 shows the current of primary segment 1 during the vehicle moves along the track. The green line shows the turn on time of the segment, and the position is between the circular orbit 0-1. The red line shows the turn off time of the segment, and the position is between the circular orbit 14-15.

Reviewer 2 Report

1- Please revise the paper for English language and style: 

- There are a number of typos and spelling mistakes, such as: "Table 3. The capacitance induction parameter ..", should be in plural form; parameters.

- There are a number of grammatical mistakes, such as: "At this time, make the side length of the transmitting coil ...". Consider using professional language which avoids imperative mood.

- Also, it is preferable to use concise technical language. Sentences like: "This method can detect whether the dynamic wireless charging electric vehicle enters or leaves the charging area by the current lag behind the voltage phase angle in the transmitting coil, so as to control whether the high-power wireless charging power supply of the charging coil in the power supply area is turned on or off according to the electric vehicle position." is too long and does not transmit a clear idea. Consider using shorter sentences.

2- The authors should include past references to help the reader understand the different formulas and derivations carried out throughout the paper, especially in Section 2: Analysis of impedance phase angle model of resonant port.

3- Section 5, regarding the experimental verification, is poor. It would be more interesting if the authors provided more pictures of the test prototype, and more insight and discussion on the results obtained.

Author Response

Dear Reviewer,

        The authors would like to thank the encouraging and insightful comments. The manuscript has been revised, considering all the comments. The salient additions to the paper were highlighted in red.

  • Reviewer: #2

(1) Please revise the paper for English language and style: - There are a number of typos and spelling mistakes, such as: "Table 3. The capacitance induction parameter ..", should be in plural form; parameters.

- There are a number of grammatical mistakes, such as: "At this time, make the side length of the transmitting coil ...". Consider using professional language which avoids imperative mood.

- Also, it is preferable to use concise technical language. Sentences like: "This method can detect whether the dynamic wireless charging electric vehicle enters or leaves the charging area by the current lag behind the voltage phase angle in the transmitting coil, so as to control whether the high-power wireless charging power supply of the charging coil in the power supply area is turned on or off according to the electric vehicle position." is too long and does not transmit a clear idea. Consider using shorter sentences.

Response: Thanks for your valuable suggestions. Now, in the revised paper, we have revised the paper for English language and style. The typos, spelling mistakes and also the grammatical mistakes have been fixed.

(2) The authors should include past references to help the reader understand the different formulas and derivations carried out throughout the paper, especially in Section 2: Analysis of impedance phase angle model of resonant port.

Response: Thank you for your suggestion. Now in the revised paper, we have added more references to help the reader understand the different formulas and derivations.

  1. X. Dai, J. Jiang, Y. Li and T. Yang, "A Phase-Shifted Control for Wireless Power Transfer System by Using Dual Excitation Units, vol. 10, pp. 1000-1015.
  2. Y. Cheng and Y. Shu, "A New Analytical Calculation of the Mutual Inductance of the Coaxial Spiral Rectangular Coils," in IEEE Transactions on Magnetics, vol. 50, no. 4, pp. 1-6
  3. Grover F. Inductance calculations New York: D. Van Nostrand Company, 1946.

(3) Section 5, regarding the experimental verification, is poor. It would be more interesting if the authors provided more pictures of the test prototype, and more insight and discussion on the results obtained.

Response: Thank you for your comment. Now in the revised paper, we have added more experimental results.

Figure 11 shows the current of primary segment 1 during the vehicle moves along the track. The green line shows the turn on time of the segment, and the position is between the circular orbit 0-1. The red line shows the turn off time of the segment, and the position is between the circular orbit 14-15.

Reviewer 3 Report

Last days the dynamic wireless charging  systems  are very useful technique, because due to the strong electromagnetic interference may be problematic. The new detection method based on phase angle sensing is proposed in this paper. The proposed method is quite robust when the parameter deviation is 4%.

Notice:

The term ZSreflect can be changed as ZPS , i.e. Z secondary recalculated into primary part in my opinion.

Proposed method is verified by Matlab simulation, results in table and figure in part 4, and experimental verification in part 5 as well.

All references are cited in the text. 

Author Response

Dear Reviewer,

        The authors would like to thank the encouraging and insightful comments. The manuscript has been revised, considering all the comments. The salient additions to the paper were highlighted in red.

  • Reviewer: #3

Comments to Author: Last days the dynamic wireless charging systems are very useful technique, because due to the strong electromagnetic interference may be problematic. The new detection method based on phase angle sensing is proposed in this paper. The proposed method is quite robust when the parameter deviation is 4%.

(1) The term ZSreflect can be changed as ZPS , i.e. Z secondary recalculated into primary part in my opinion.

Response: Thanks for your valuable suggestions. Now in the revised paper, we have changed ZSreflect to ZPS.

Mapping impedance refers to the equivalent impedance when converting the impedance of the secondary circuit to the primary circuit, set to ZPS, and there are:

(2)Proposed method is verified by Matlab simulation, results in table and figure in part 4, and experimental verification in part 5 as well. All references are cited in the text.

Response: Thank you for your approval.

Reviewer 4 Report

General

The paper in its present form presents many weak points. The analysis stops at a preliminary level. No adequate reference to past works on the topic is made.

Important aspects are not assessed in the paper:
- how the ZVS is guaranteed according to the changes in the operating frequency required by the proposed technique?

- as widely analyzed in DOI 10.1109/ACCESS.2020.3025052 , the efficiency and the transmitted power are strongly influenced by changes in the operating frequency. These aspects should be validated at least in the experimental verification.

Introduction

In which way WPT systems are “high intelligent”? What did the authors mean?

Please, consider that are not the WPT charging systems that require a high-capacity battery: this quantity depends on the desired range and is independent of the recharging technology! In principle, the same statement could be applied to standard AC plug-in charging systems… The following statements should be revised accordingly.

Please, consider that dynamic WPT systems are not always intended to recharge the vehicle battery but instead provide the power instantly required for the movement. The charge only occurs, at low speeds, if the power received by the vehicle is lower than the power required at the vehicle wheels.

In relation to the use of sensors for coil activation in dynamic WPT systems, consider previous works, such as DOI 10.1109/TIE.2018.2803719, which developed sensorless techniques. It should be fair to mention this as other previous work that dealt with the same topic dealt with in the manuscript. Moreover, the suggested work seems to propose a similar solution but with a lower count of components: in which way does the proposed solution differ from this one or improve the solution?

The description of the system done in the introduction and referred to Fig. 1 is too raw and synthetic. The whole system and concept should be better explained.

Section III

The analytical calculation of self and mutual inductance for rectangular coils has been effectively presented in many papers. Among the most relevant there is the following DOI 10.1016/j.apenergy.2008.05.009

However, the analytical formulation has very limited applicability and is ineffective in all real applications in which ferromagnetic and conductive materials are adopted. Moreover, the actual way of calculation adopted for the mutual inductance is not clear. It seems based on a superposition of the effect of the different turns composing the coil: the authors mention a simulation-based approach but details are missing.

Section IV

kc is not defined.

Please, consider the following sentence:” In fact, when the resonant frequency of the system shifts, it can be considered that the resonant frequency is fixed and the working frequency changes.” It appears difficult to understand…

Section V

In which way the design parameters of the coils have been selected? This is completely unclear.

Why is the coil resistance referred to as “parasitic”? It is an intrinsic characteristic of all conductors and it is not a parasitic quantity…

In table 2 it is mentioned a “super capacitance value of energy storage module” but no mention is made about this component and its use in the text.

The content of fig. 10 and fig. 11 are not properly introduced and discussed either in the text or in the figures’ caption.

Author Response

Dear Reviewer,

        The authors would like to thank the encouraging and insightful comments. The manuscript has been revised, considering all the comments. The salient additions to the paper were highlighted in red. Please find the detailed response  in the attachment.

Round 2

Reviewer 4 Report

The authors addressed all the concerns of the reviewer. Only minor concerns remain as detailed in the following.

Generally, the unit of measurement must be properly spaced from the related digit.

In the abstract, the threshold value indicated in the last sentence (300) should have a unit of measurement.

Concerning the added description of the system, it seems that Li represents also the self-inductance of the coil. Hence, I suggest modifying the related sentences as “Each charging area contains a wireless charging transmitting coil having/with/of self-inductance Li” and “The electric vehicle is equipped with receiving coil having/with/of self-inductance Ls”.

The authors modified “parasitic resistance” with “internal resistance” I suggest simply using the expression “coil resistance/resistance of the coil”. This is because it is unclear what the meaning of “internal” resistance should be: if internal resistance is made explicit, what should the respective “external” resistance be?

Figures 10 and 12 could be increased to improve readability.

Author Response

(The authors gave the same response as above.)
